# Gender-Related Differences in BMP Expression and Adult Hippocampal Neurogenesis within Joint-Hippocampal Axis in a Rat Model of Rheumatoid Arthritis

**DOI:** 10.3390/ijms222212163

**Published:** 2021-11-10

**Authors:** Hrvoje Omrčen, Sanja Zoričić Cvek, Lara Batičić, Sandra Šućurović, Tanja Grubić Kezele

**Affiliations:** 1Department of Anatomy, Faculty of Medicine, University of Rijeka, Braće Branchetta 20, 51000 Rijeka, Croatia; hrvoje.omrcen@medri.uniri.hr (H.O.); sanja.zoricic@medri.uniri.hr (S.Z.C.); 2Department of Medical Chemistry, Biochemistry and Clinical Chemistry, Faculty of Medicine, University of Rijeka, Braće Branchetta 20, 51000 Rijeka, Croatia; lara.baticic@medri.uniri.hr; 3Department of Hematology, Medical Centre Ljubljana, Zaloška 7, 1000 Ljubljana, Slovenia; sandra.sucurovic@kclj.si; 4Department of Physiology, Immunology and Pathophysiology, Faculty of Medicine, University of Rijeka, Braće Branchetta 20, 51000 Rijeka, Croatia; 5Clinical Department for Clinical Microbiology, Clinical Hospital Centre Rijeka, Krešimirova 42, 51000 Rijeka, Croatia

**Keywords:** bone morphogenetic proteins, rheumatoid arthritis, interleukin-17, tumor necrosis factor-alpha, hippocampus, adult neurogenesis, Noggin, Gremlin

## Abstract

BMPs regulate synovial quiescence and adult neurogenesis in the hippocampus in non-stress conditions. However, changes in BMP expression that are induced by inflammation during rheumatoid arthritis (RA) have not yet been reported. Here, we show that signalling with synovial BMPs (BMP-4 and -7) mediates the effect of systemic inflammation on adult neurogenesis in the hippocampus during pristane-induced arthritis (PIA) in *Dark Agouti* (DA) rats, an animal model of RA. Moreover, we show gender differences in BMP expressions and their antagonists (Noggin and Gremlin) during PIA and their correlations with the clinical course and IL-17A and TNF-α levels in serum. Our results indicate gender differences in the clinical course, where male rats showed earlier onset and earlier recovery but a worse clinical course in the first two phases of the disease (onset and peak), which correlates with the initial increase of serum IL-17A level. The clinical course of the female rats worsened in remission. Their prolonged symptoms could be a reflection of an increased TNF-α level in serum during remission. Synovial inflammation was greater in females in PIA-remission with greater synovial BMP and antagonist expressions. More significant correlations between serum cytokines (IL-17A and TNF-α), and synovial BMPs and their antagonists were found in females than in males. On the other hand, males showed an increase in hippocampal BMP-4 expression during the acute phase, but both genders showed a decrease in antagonist expressions during PIA in general. Both genders showed a decrease in the number of Ki-67^+^ and SOX-2^+^ and DCX^+^ cells and in the ratio of DCX^+^ to Ki67^+^ cells in the dentate gyrus during PIA. However, in PIA remission, females showed a faster increase in the number of Ki67^+^, SOX-2^+^, and DCX^+^ cells and a faster increase in the DCX/Ki67 ratio than males. Both genders showed an increase of hippocampal BMP-7 expression during remission, although males constantly showed greater BMP-7 expression at all time points. Our data show that gender differences exist in the BMP expressions in the periphery–hippocampus axis and in the IL-17A and TNF-α levels in serum, which could imply differences in the mechanisms for the onset and progression of the disease, the clinical course severity, and adult neurogenesis with subsequent neurological complications between genders.

## 1. Introduction

Rheumatoid arthritis (RA) is a chronic autoimmune disease that is characterized by the synovial inflammation and deformation of joints and adjacent bones. The pathogenesis of RA is driven by a complex interplay between the proinflammatory pathways that bring about joint and systemic inflammation [1]. It is still an unexplored mystery as to how the intimate crosstalk between the peripheral and central immune responses influence each other. Beyond joint pathology and difficulties in physical functioning, RA is often associated with a neuropsychiatric comorbidity, including depression, anxiety, cognitive deficits, and pain, which substantially affect the daily quality of life of the patients suffering from this disease [2]. 

Depression negatively affects the ability of RA patients to function while enduring physical symptoms, such as pain and fatigue [3,4]. Long-term exposure to raised pro-inflammatory cytokines can cause maladaptive responses to sickness behaviour, causing fatigue, pain, fever, anhedonia, and depression [5]. Observational studies have described a high prevalence of depression and anxiety in RA [6], and local and systemic inflammation plays an important role [7].

After receiving inflammatory signals from the periphery during RA, resident cells in the central nervous system (CNS), particularly in the microglia and astrocytes, are able to acquire an activated phenotype and maintain a neuroinflammatory state.

The underlying pathogenesis by which neuroinflammation results in these neuropsychiatric symptoms and disorders is impaired neuronal function that includes several mechanisms: alterations in neurotransmitter signaling, dynamic modulation of dendritic spines and neuronal networks, and impaired adult hippocampal neurogenesis [1,8]. Adult hippocampal neurogenesis, the generation of new neurons in the dentate gyrus throughout a person’s lifetime, is a physiological process that contributes to learning, memory, pattern separation, and emotions [9].

Changes in the hippocampal neurogenic niche during RA depend on genetic and epigenetic factors that ensure cellular and environmental homeostasis and regulate the interactions of immunocompetent neural cells in order to provide neurostability [10].

Moreover, an increase of proinflammatory factors and cytokines (e.g., TNF-α and IL-17A) and a reduction of neurotrophic factors have been reported to modulate hippocampal neurogenesis and neuroplasticity in chronic pain and depression in chronic inflammatory diseases such as RA [11]. Inflammation is detrimental for hippocampal neurogenesis in the adult brain and has subsequent consequences, especially in chronic inflammatory diseases such as RA [12,13,14]. Substantial progress has been made in identifying the extracellular signalling pathways that regulate neural stem cell (NSC) and precursor cell biology in the hippocampus. NSCs in the adult hippocampus divide infrequently, and the molecules that modulate their quiescence are largely unknown. Bone morphogenetic proteins (BMPs) are anabolic candidates with pleiotropic functions in the development, homeostasis, and repair of various tissues [15]. Among the signalling pathways that govern stem cells, BMPs are critical regulators of stem cell self-renewing divisions and maintenance in a wide variety of niches, including the hippocampal one [16,17]. In the adult brain, the contribution of BMP signalling to stem cell regulation remains largely unexplored, although recently, it has been suggested that the BMP pathway may have a central role in modulating NSC function [18,19,20,21]. A substantial number of studies explain that BMP signalling differently affects cell proliferation and neurogenesis [22]. 

Furthermore, BMP signalling has been proposed to be involved in cartilage and bone repair in arthritis. Moreover, they may be equally important in modulating synoviocytes in inflamed synovium as well [23]. However, their role in degenerative joint diseases is still insufficiently understood. Although we have substantial information about the pathophysiology of the disease, with various groups of immune cells and soluble mediators having been identified to participate in the pathogenesis, several aspects of the altered immune functions and regulation in RA remain controversial. Animal models are especially useful in such scenarios. A single intradermal injection of the mineral oil pristane in susceptible *Dark Agouti* (DA) rats induces erosive arthritis that closely mimics RA [24].

Furthermore, it is well described that RA affects women more often than men, where the gender ratio is typically around 3:1 [25]. The reasons for this overrepresentation of women are not clear, but genetic (X-linked) factors and hormonal aspects are likely to be involved [26]. However, the available literature is unclear about gender differences in the *RA* disease course and the underlying mechanisms of the disease, including BMP expression. 

This paper will concentrate on gender differences in BMP expression within the periphery–hippocampal axis and its relationship with the clinical course and main inflammatory RA cytokines, TNF-α and IL-17A, that compromise adult hippocampal neurogenesis and probably contribute to neuropsychiatric disorders in RA. We hypothesize that gender differences exist in BMP expression and in the serum levels of TNF- α and IL-17A, which compromises adult hippocampal neurogenesis in RA, a key regulator of normal neuropsychiatric functioning. In addition, this work will be the basis for exploring new mechanisms in future experiments, including the relationship with the sex hormones that are implicated in RA pathogenesis.

## 2. Materials and Methods

### 2.1. Experimental Animals

Experiments were performed on 7–8-week-old male and female DA rats. Animals were housed under standard light, temperature, and humidity conditions with unlimited access to food and water. Experimental procedures were in compliance with Croatian laws and regulations (OG 135/06; OG 37/13; OG 125/13; OG 055/2013) and with the guidelines set by the European Community Council Directive (86/609/EEC). The experimental protocol was approved by the Croatian Ministry of Agriculture (525-10/0255-19-5) and the Ethics Committee of the University of Rijeka, Faculty of Medicine (003-08/21-01/38).

### 2.2. Arthritis Induction and Disease Course Evaluation

Pristane-induced arthritis (PIA) was induced by an intradermal injection of 150 μL of pristane (2,6,10,14-tetramethylpentadecane, ≥ 98%, Sigma-Aldrich, St. Louis, MD, USA) at the dorsal side of the tail base. Control rats were intradermally injected with saline. Arthritis development was monitored by a blinded observer in all four limbs using a semiquantitative scoring system [27]. Rats were scored every day after injection. Briefly, one point was given for each swollen or red finger/toe (interphalangeal joints), one point was given for each swollen or red metacarpophalangeal/metatarsophalangeal joint, and five points were given for a swollen wrist/ankle, depending on severity (the maximum score per limb and rat was 15 and 60, respectively). Scores were not given for deformations if they were not accompanied by erythema or swelling. The day of disease remission is defined here as the first of at least three consecutive scoring days with declining arthritis scores. 

### 2.3. Experimental Design

Both the female and male rats were randomly divided into two main groups: the control group (N = 20: females n = 10, males n = 10) and the experimental group (rats with induced PIA) (N = 120: females n = 60, males n = 60). PIA rats were further divided into three subgroups that were sacrificed at different time points after PIA induction, i.e., on the day when the first symptoms appeared between 9th and 12th day (PIA onset), on the peak of disease activity between 16th and 20th day (PIA peak) and in the remission phase between 20th and 25th day (PIA remission). The control group was treated with physiological saline solution and was sacrificed after a few days. Rats were sacrificed by exsanguination in deep anaesthesia, which was induced by a combination of ketamine (80 mg/kg) and xylazine (5 mg/kg) that were administered by intraperitoneal injection, as previously described [28,29]. All experiments were performed according to the guidance of the European Community Council Directive (86/609/EEC) and the recommendations of the National Centre for the Replacement, Refinement and Reduction of Animals in Research.

### 2.4. Tissue Preparation for Paraffin Slices

#### 2.4.1. The Rat Brain 

Rat brain hemisphere samples were fixed in 4% buffered paraformaldehyde (Sigma-Aldrich, St. Louis, MD, USA) solution for 24 h. The tissue was then embedded in paraffin wax, and sections were cut at 4 μm using the HM340E microtome (Microtome, Walldorf, Germany). 

#### 2.4.2. Hind Paws 

Hind paws from rats were collected and fixed in 4% buffered paraformaldehyde (Sigma-Aldrich, St. Louis, MD, USA) solution for 72 h, after which the specimens were immersed in 70% ethanol for 20 min, and the hind paws were then decalcified using a commercial decalcifying solution (Osteofast 2, Biognost, Zagreb, Croatia) for 12 h with the solution being changed every 6 h. The decalcification end point was determined by puncturing the specimen with a very thin needle (27G). Once decalcified, the specimens were washed in three changes of PBS (20 min each), followed by paraffin embedding. Serial paraffin-embedded tissue sections of the hind paws were cut to a 4 μm thickness and were incubated for 12 h under 37 °C until the samples were completely dry.

### 2.5. Histological, Immunohistochemical and Immunofluorescence Staining

#### 2.5.1. Histochemistry

For orientation, the rat hind paw slides were stained with hematoxylin and eosin. After deparaffinization in two changes of xylene and rehydration in graded ethanol, slides were stained with Instant Hematoxylin (Thermo Shandon, Pittsburgh, PA, USA) for 5 min, washed under running tap water, and then counterstained with Instant Eosin-Y (Thermo Shandon, Pittsburgh, PA, USA) for 10 min. Slides were rinsed in distilled water, dehydrated in graded ethanol, and mounted with Entellan (Sigma-Aldrich, St. Louis, MD, USA). 

#### 2.5.2. Immunohistochemistry

Immunohistochemical labelling of BMP-4, BMP-7, Noggin, Gremlin, and CD3 was performed on paraffin-embedded tissues. After deparaffinization and rehydration in graded ethanol, slides were subjected to heat antigen retrieval with citrate buffer (0.01 M sodium citrate pH 6.0) for 10 min at 90 °C, and the slides were then cooled under running tap water. Slices were then incubated with 3% BSA for 15 min to eliminate background staining.

After rinsing in PBS, goat polyclonal IgG anti-BMP-4 (sc-12721), anti-BMP-7 (sc-34766), anti-Noggin (sc-16627), anti-Gremlin (sc-18274) antibodies (Santa Cruz Biotechnology, Dallas, TX, USA, diluted 1:100 with 1% BSA in PBS) and rabbit polyclonal IgG anti-CD3 (Abcam, Cambridge, MA, USA, ab5690, diluted 1:100 with 1% BSA in PBS) were added to the tissue samples and were incubated overnight at 4 °C in a humid environment. The next day, the slides were treated with Dako REAL Peroxidase-Blocking Solution (DAKO, Glostrup, Denmark) for 10 min to block endogenous peroxidase activity. Slides were then rinsed in PBS and were treated with Streptavidin HRP Conjugate (Invitrogen, Rockford, IL, USA) for 15 min. After rinsing, the slides were incubated with secondary antibodies for 1 h.

The immunoreaction products were visualized by adding substrate chromogen (DAB) solution. Slides were counterstained with hematoxylin, dehydrated through graded alcohol, and mounted using Entellan (Sigma-Aldrich, St. Louis, MD, USA). The staining specificity was confirmed with negative controls. Tissue samples were treated with an identical procedure under the same conditions but with the omission of polyclonal primary antibodies. The photomicrographs were taken and examined under an Olympus BX51 light microscope equipped with an Olympus DP70 camera (Olympus, Tokyo, Japan).

#### 2.5.3. Immunofluorescence

Immunofluorescence labelling was also performed on paraffin-embedded tissue sections. Nonspecific binding was blocked by one-hour incubation with 1% BSA in PBS containing 0.001% NaN_3_ at room temperature, as previously described [30]. The following primary antibodies were used: goat polyclonal IgG anti-BMP-4, anti-BMP-7, anti-Noggin, anti-Gremlin antibodies, mouse monoclonal anti-SOX-2 IgG2a antibody (Santa Cruz Biotechnology, Dallas, TX, USA, diluted 1:100), rabbit polyclonal IgG anti-doublecortin (anti-DCX), and anti-Ki67 antibodies (Abcam, Cambridge, UK, 1:1000 and 1:200). Primary antibodies were diluted in blocking solution and were incubated with tissue sections overnight at 4 °C in a humid environment. For the visualisation of the immunocomplexes, the following secondary antibodies were used: Alexa Fluor donkey anti-rabbit IgG 594 nm (Molecular Probes, Carlsbad, CA, USA, 1:500) and Alexa Fluor donkey anti-goat IgG 488 nm (Molecular Probes, Carlsbad, CA, USA, 1:300). Secondary antibodies were diluted in blocking solution and were incubated with tissue sections in the dark for 1 h at room temperature in a humid environment. Nuclei were visualized by DAPI staining (1:1000 in PBS for 5 min; Molecular Probes, Carlsbad, CA, USA). Afterwards, the slides were washed in PBS and were mounted with Mowiol (Sigma-Aldrich, Germany). The photomicrographs were taken under a fluorescent microscope equipped with a DP71CCD camera (Olympus, Japan) and Cell F imaging software. Adult hippocampal neurogenesis was assessed by the number of proliferating cells (Ki67^+^ cells), the number of neural progenitors (SOX-2^+^ cells), neuroblasts/immature neurons (DCX^+^ cells), and the ratio of DCX^+^ to Ki67^+^ cells, reflecting hippocampal proliferation as well as newly born cell maturation speed and survival. The specificity of the reaction was confirmed by the omission of the primary antibodies on the slides that had been treated with an identical procedure under the same conditions. The photomicrographs were taken and examined under an Olympus BX51 light microscope equipped with an Olympus DP70 camera (Olympus, Japan).

### 2.6. Immunohistochemical/Fluorescence Staining Quantification, Cell Counting, and Histopathological Evaluation of Joint Inflammation

#### 2.6.1. Quantification

The immunohistochemical/fluorescence staining quantification of protein expression was performed on 4 μm tissue sections from paraffin-embedded tissues of the brain and hind paws using Cell F v3.1 software (Olympus Soft Imaging Solutions), as previously described [28]. Captured images were subjected to intensity separation. They were subsequently inverted, resulting in grayscale images with different intensity ranges, depending on the strength of immunohistochemical staining or fluorescence signals. Regions of interest (ROIs) were arranged to cover the analysed area, and mean grey values were measured. ROI surface size was always equal for each analyzed area. Twelve ROIs were analyzed per field (400×) on three separate microscopic slides of different tissue samples per animal, which were obtained from six animals per group. The data were expressed as the mean grey value ± SD.

#### 2.6.2. Cell Counting 

We estimated the DCX^+^, SOX^+^, and Ki67^+^ cells in the dentate gyrus using antibodies and DAPI staining, respectively. DCX^+^, SOX^+^, and Ki67^+^ cells were counted manually in an image surface area of 0.053 mm^2^ and at a magnification of 400×. Results were expressed as a mean number of cells per mm^2^ [28,31].

#### 2.6.3. Histopathological Evaluation of Synovitis

Synovial inflammation was scored with the semiquantitative Krenn synovitis scoring system, which examines the synovial lining layer cell hyperplasia, stromal cells activation, and inflammatory infiltrate using routine hematoxylin and eosin- (H&E) stained slides. All components were graded from 0 to 3, with a maximum score of 9 [32]. Low-grade synovitis was defined by a cumulative score between 2 and 4, while high-grade synovitis was defined by a cumulative score higher than 5.

### 2.7. Determination of Serum Concentrations of IL-17A and TNF-α

The quantification of interleukin IL-17A and TNF-α levels in serum in the defined rat groups were determined quantitatively using commercially available ELISA kits (Invitrogen, ThermoFisher Scientific). Analytical procedures were performed according to the manufacturer’s instructions. All measurements were made in duplicates. According to the standards, readouts and calculations of concentrations were performed using a microplate reader (Bio-Tek EL808, Winooski, VT, USA) and the Bio-Tek Instruments Inc. software, EL808. Results are expressed as pg/mL of serum (mean ± SD). 

### 2.8. Statistical Analysis

The data were evaluated with Statistica, version 13 (TIBCO Software Inc., 2017, Palo Alto, CA, USA). Data distribution was tested for normality using the Kolmogorov–Smirnov test. Differences between genders and groups were assessed with either one-way analysis of variance (ANOVA) followed by the post hoc Scheffé test or the Mann–Whitney *U* test. The Pearson correlation (*r*) was used to determine the association between adult neurogenesis (Ki67^+,^ DCX^+^ cells, and DCX/Ki67 ratio) and BMPs, their antagonists, and TNF-α and IL-17A levels in serum. The data were expressed as mean ± SD, and the significance level was set at *p* < 0.05.

## 3. Results

PIA was induced in genetically susceptible DA rats with an incidence of 99.2% (119 out of 120 animals) in both genders. Rats who had been immunized with pristane showed normal weight gain before the onset of the disease (data not shown). The decline in body weight that followed the onset of arthritis was proportional to disease severity. As the remission phase started, the rats began to gain weight again. The clinical course and the expression profiles of BMP-4, BMP-7, Noggin, Gremlin, and CD3 were examined in the hippocampus/dentate gyrus and in the metatarsophalangeal joints of the hind paws. Furthermore, expressions of Ki-67, DCX, and SOX-2 were examined in the hippocampus/dentate gyrus. Concentrations of IL-17A and TNF-α were evaluated from blood serum.

### 3.1. Clinical Course 

The PIA subgroups of rats were monitored daily over the course of 30 days. Male rats showed an earlier onset of PIA (Figure 1), with a maximum mean score of 28.2 ± 1.9 when approaching the peak of the disease. The maximum mean score in the female PIA group upon reaching the peak of the disease was 27.0 ± 6.1. However, the remission phase began earlier in the male rats, and the female rats processed the remission phase with a higher level of clinical scores, which is consistent with a much higher TNF-α level in serum in females (Figure 5A and Table 1).

### 3.2. Adult Hippocampal Neurogenesis

PIA decreased the number of proliferating cells (Ki67^+^), neural progenitors (SOX-2^+^ cells), neuroblasts/immature neurons (DCX^+^ cells), and the DCX/Ki67 ratio in both genders (Figure 2). However, the proliferation of hippocampal/dentate gyrus cells (Ki67^+^) in the male rats initially increased with the first signs of PIA, which is consistent with the initial increase in IL-17A (Table 1) in the serum and BMP-4 in the hippocampus (Figure 3B). This isolated increase in cell proliferation (Ki67^+^) or in stem/progenitor cell proliferation (SOX-2^+^) without further differentiation into young neurons (DCX^+^ cells) indicates that some factors (e.g., IL-17A and BMP-4) are probably limiting neuronal cell differentiation and survival, thus affecting adult neurogenesis.

**Figure 1 ijms-22-12163-f001:**
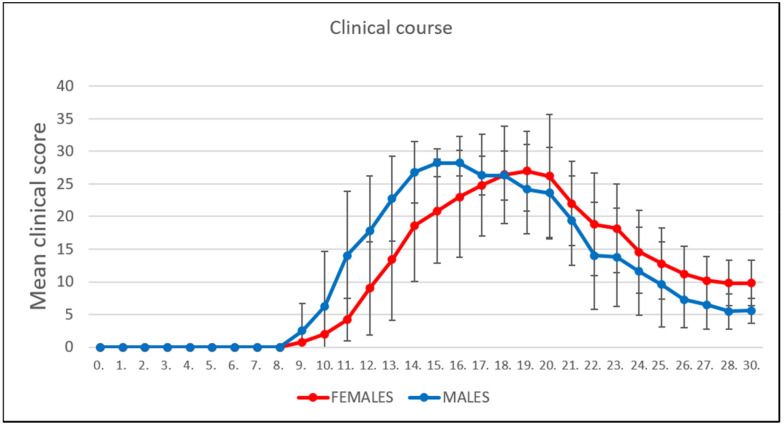
Clinical course. The clinical course in the male (n = 45) and female (n = 45) rat groups. Values are presented as mean ± SD (Mann–Whitney U test) using PIA scores of each animal for every day.

**Table 1 ijms-22-12163-t001:** Pearson’s correlation between hippocampal neurogenesis, hippocampal/dentate gyrus expressions of BMPs, Noggin, Gremlin, and IL-17A and TNF-α levels in serum.

Variable	Gender	Average Grey Value BMP4	Average Grey Value BMP7	Average Grey Value Noggin	Average Grey Value Gremlin	TNF-α	IL-17A
DCX^+^ cells	Males	0.01	−0.22	0.83 ***	0.10	−0.13	−0.31
	Females	0.82 ***	0.47	0.62 *	0.02	0.54 *	−0.59 *
Ki67^+^ cells	Males	0.37	−0.45	0.31	0.69 **	−0.01	0.44
	Females	0.82 ***	0.45	0.57 *	−0.22	0.51 *	−0.57 *
DCX/Ki67	Males	−0.27	0.22	0.56 *	−0.45	−0.19	−0.63 *
	Females	0.35	0.18	0.33	0.52 *	0.23	−0.23
TNF-α	Males	−0.42	0.44	−0.11	−0.13	/	/
	Females	0.15	0.96 ***	−0.19	−0.11	/	/
IL-17A	Males	0.54 *	−0.28	−0.65 *	0.85 ***	/	/
	Females	−0.79 ***	0.0002	−0.82 ***	0.05	/	/

* *p* < 0.05, ** *p* < 0.01 and *** *p* < 0.001.

In the remission phase of the disease (PIA remission), the female rats showed a faster increase in the number of Ki67^+^ cells (Figure 2C), SOX-2^+^ cells (Figure 2E: *p* = 0.039), in the number of DCX^+^ cells (Figure 2B: *p* = 0.002), and in the DCX/Ki67 ratio (Figure 2D) than male rats.

**Figure 2 ijms-22-12163-f002:**
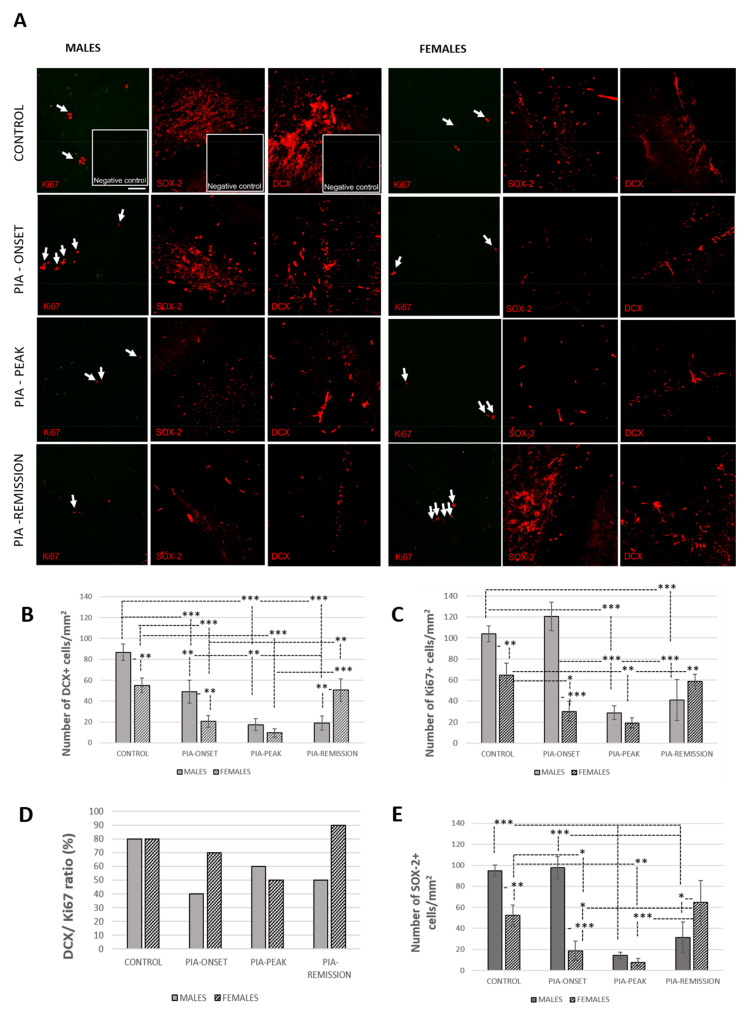
Pristane-induced arthritis downregulates the number of Ki67^+^, DCX^+^, and SOX-2^+^ cells as well as the DCX/Ki67 ratio in DA rats. (**A**) Representative immunofluorescent pictures show staining with anti-Ki67, anti-DCX, and anti-SOX-2 antibodies in paraffin-embedded sections of brain tissue obtained from male and female DA rats: control (treated with saline); PIA onset (between 9th and 12th day after induction); PIA peak (between 16th and 20th day after induction); PIA remission (between 20th and 25th day after induction). Arrows: Ki67^+^ cells. (**B**,**C**,**E**) DCX, Ki67, and SOX-2 immunoreactivities in the hippocampus/dentate gyrus. The number of DCX^+^, Ki67^+^, and SOX-2^+^ cells per mm^2^ was manually counted in regions of interest (12 ROI/4 μm slide x 3 slides/rat × 6 rats/group). Values are expressed as mean value ± SD (*N* = 24). One-way ANOVA followed by the post hoc Scheffé test: * *p* < 0.05, ** *p* < 0.01, and *** *p* < 0.001. Inserts show staining in slides incubated without primary anti-Ki67, anti-SOX-2, and anti-DCX antibodies (negative controls). Scale bars indicate 50 μm. (**D**) The ratio of DCX^+^ to Ki67^+^ cells is expressed as a percentage (%).

**Figure 3 ijms-22-12163-f003:**
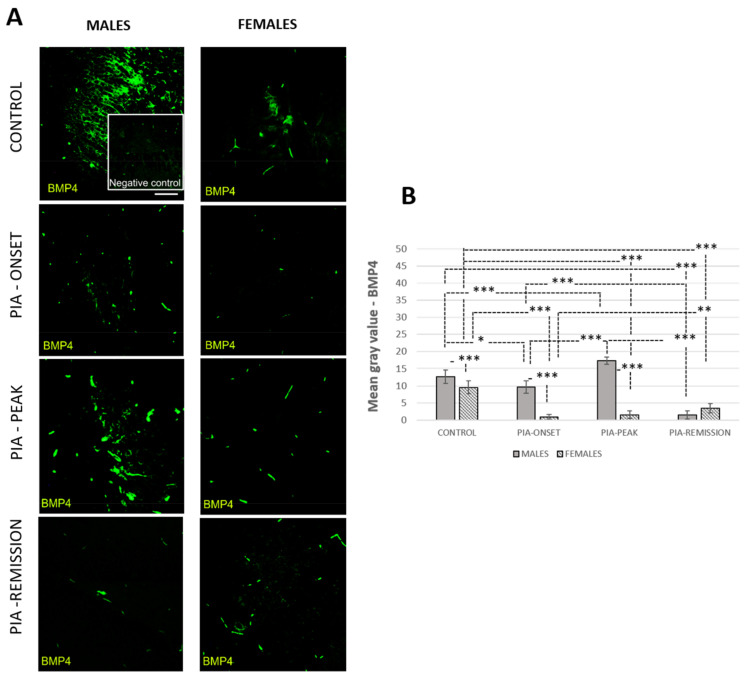
Pristane-induced arthritis upregulates BMP-4 expression in the hippocampus/dentate gyrus in male rats. (**A**) Representative immunofluorescent pictures show staining with anti-BMP-4 antibody in paraffin-embedded sections of brain tissue obtained from male and female DA rats: control (treated with saline); PIA onset (between 9th and 12th day after induction); PIA peak (between 16th and 20th day after induction); PIA remission (between 20th and 25th day after induction). (**B**) BMP-4 immunoreactivity in the hippocampus/dentate gyrus. The immunofluorescent signal quantification was performed using Cell F v3.1 software analysis on 12 regions of interest (3 slides/rat × six animals/group). Values are expressed as mean grey value ± SD (*N* = 24). One-way ANOVA followed by the post hoc Scheffé test: * *p* < 0.05, ** *p* < 0.01, and *** *p* < 0.001. Insert shows staining in a slide incubated without primary anti-BMP4 antibody (negative control). Scale bars indicate 50 μm.

### 3.3. Male Rats Express a Higher Level of Hippocampal/Dentate Gyrus BMP-4 Than Female Rats

The profiling of hippocampal/dentate gyrus BMP-4 by immunofluorescence showed significant upregulation during the acute phase (onset and peak) of the disease in male rats compared to the findings in female rats (Figure 3B: PIA-onset *p* < 0.001; PIA-peak *p* < 0.001).

This is consistent with more progressive clinical course onset in males and correlates with the increased IL-17A level in the serum (Table 1: r = 0.54, *p* = 0.040). Data also show that the untreated control male rats possessed a higher level of BMP-4 and its antagonist Noggin (Appendix A) than female rats, which correlates with a larger number of neuroblasts/immature neurons (DCX^+^ cells) and the ratio of DCX^+^ to Ki-67^+^ cells (DCX/Ki67), reflecting a higher basic level of hippocampal proliferation as well as newly born cell maturation speed and survival (Table 1: r = 0.83, *p* < 0.001; r = 56, *p* = 0.030). These results may imply a higher level of protection for adult neurogenesis in general in male rats.

### 3.4. Male Rats Express a Higher Level of Hippocampal/Dentate Gyrus BMP-7 Than Female Rats during PIA

The profiling of hippocampal/dentate gyrus BMP-7 by immunofluorescence showed significant downregulation during the acute phase (onset and peak) in males (*p* < 0.001) (Figure 4). However, the level of BMP-7 was higher in the control and PIA peak in males than they were in females (*p* < 0.001) (Figure 4B). In the remission phase, BMP-7 was significantly upregulated in both genders (*p* < 0.001), even above the level of control animals (Figure 4B). These findings correlate with TNF-α level in serum, especially in female rats (Table 1: r = 0.96, *p* < 0.001). We assume that this could be related to a worse prognosis and chronicity in females. In general, Gremlin had a low level of expression in both genders (Appendix A), although results showed a slight increase at the onset of clinical signs in male rats (Figure 4), which is correlated with the number of proliferating (Ki67^+^) cells (r = 0.69, *p* = 0.005) and the IL-17A level in the serum (Table 1: r = 0.85, *p* = 0.001). These findings could relate to a better prognosis in males concerning adult neurogenesis.

### 3.5. TNF-α and IL-17A Levels in Serum

Results showed an increase of TNF-α in serum in female rats in the remission phase of the disease (PIA remission) compared to males (*p* < 0.001) (Figure 5A). This finding could relate to a worse prognosis and chronicity in females. However, evaluation of IL-17A in serum showed increased values in male rats at the onset of the disease compared to females (*p* < 0.001) (Figure 5B). This finding is consistent with the more progressive onset of PIA in male rats (Figure 1). Furthermore, the results showed a negative correlation of IL-17A with the Ki67^+^ and DCX^+^ cell count in females (Table 1: r = −0.57, *p* = 0.033; r = 0.59, *p* = 0.020) as well as with the DCX/Ki67 ratio in males (Table 1: r = −0.63, *p* = 0.015). In addition, the results showed that IL-17A has a positive correlation with BMP-4 expression in males (Table 1: r = 0.54, *p* = 0.040), a negative correlation with BMP-4 expression in females (Table 1: r = −0.79, *p* < 0.001), and a negative correlation with Noggin expression in both genders (Table 1: males, r = −0.65, *p* = 0.010; females, r = −0.82, *p* < 0.001). All together, these findings suggest a negative impact of IL-17A on adult neurogenesis in both genders, although a lower level of BMP-4 in females may indicate a faster recovery of adult neurogenesis (Figure 2).

### 3.6. Correlations of ADULT hippocampal Neurogenesis with Hippocampal BMPs, Noggin, Gremlin, and IL-17A and TNF-α in Serum

The hippocampal/dentate gyrus BMP-4 expression strongly correlates with the number of DCX^+^ and Ki67^+^ cells in the female dentate gyrus (r = 0.82, *p* < 0.001). However, BMP-4 expression was lower in females than in males in general, especially in acute PIA (*p* < 0.001), which negatively correlates with IL-17A level in serum (r = −0.79, *p* < 0.001).

On the contrary, BMP-4 expression in males increased during PIA and decreased again in remission, which was positively correlated with the IL-17A level in the serum (r = 0.54, *p* = 0.040). The IL-17A level in serum negatively correlates with the number of DCX^+^ and Ki67^+^ cells in the females (r = −0.59, *p* = 0.020; r = −0.57, *p* = 0.033) and with the DCX/Ki67 ratio in the males (r = −0.63, *p* = 0.015). Furthermore, the expression of antagonist Noggin was positively correlated with the number of DCX^+^ and Ki67^+^ cells in the females (r = 0.62, *p* = 0.018; r = 0.57, *p* = 0.33) and with the number of DCX^+^ cells and with DCX/Ki67 ratio in the males (r = 0.83, *p* < 0.001; r = 0.56, *p* = 0.030). In addition, the IL-17A level in the serum was negatively correlated with Noggin expression in the hippocampus/dentate gyrus in both genders (r = −0.65, *p* = 0.010; r = −0.82, *p* < 0.001). Noggin expression was downregulated during PIA in both genders as well.

**Figure 4 ijms-22-12163-f004:**
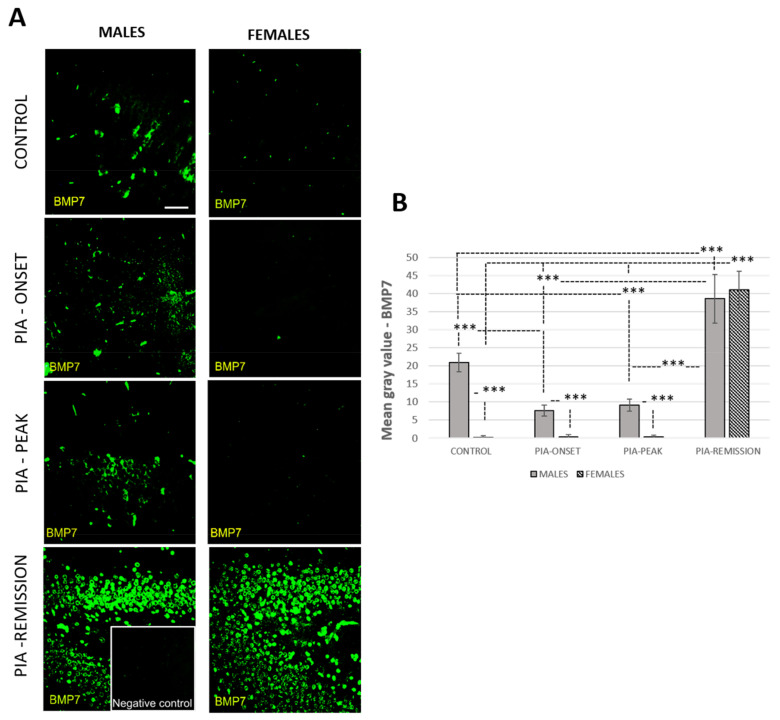
Pristane-induced arthritis upregulates BMP-7 expression in the hippocampus/dentate gyrus in male rats. (**A**) Representative immunofluorescent pictures show staining with anti-BMP-7 antibody in paraffin-embedded sections of brain tissue obtained from male and female DA rats: control (treated with saline); PIA onset (between 9th and 12th day after induction); PIA peak (between 16th and 20th day after induction); PIA remission (between 20th and 25th day after induction). (**B**) BMP-7 immunoreactivity in the hippocampus/dentate gyrus. The immunofluorescent signal quantification was performed using Cell F v3.1 software analysis on 12 regions of interest (3 slides/rat × six animals/group). Values are expressed as mean grey value ± SD (*N* = 24). One-way ANOVA followed by the post hoc Scheffé test: *** *p* < 0.001. Insert shows staining in a slide incubated without primary anti-BMP7 antibody (negative control). Scale bars indicate 50 μm.

**Figure 5 ijms-22-12163-f005:**
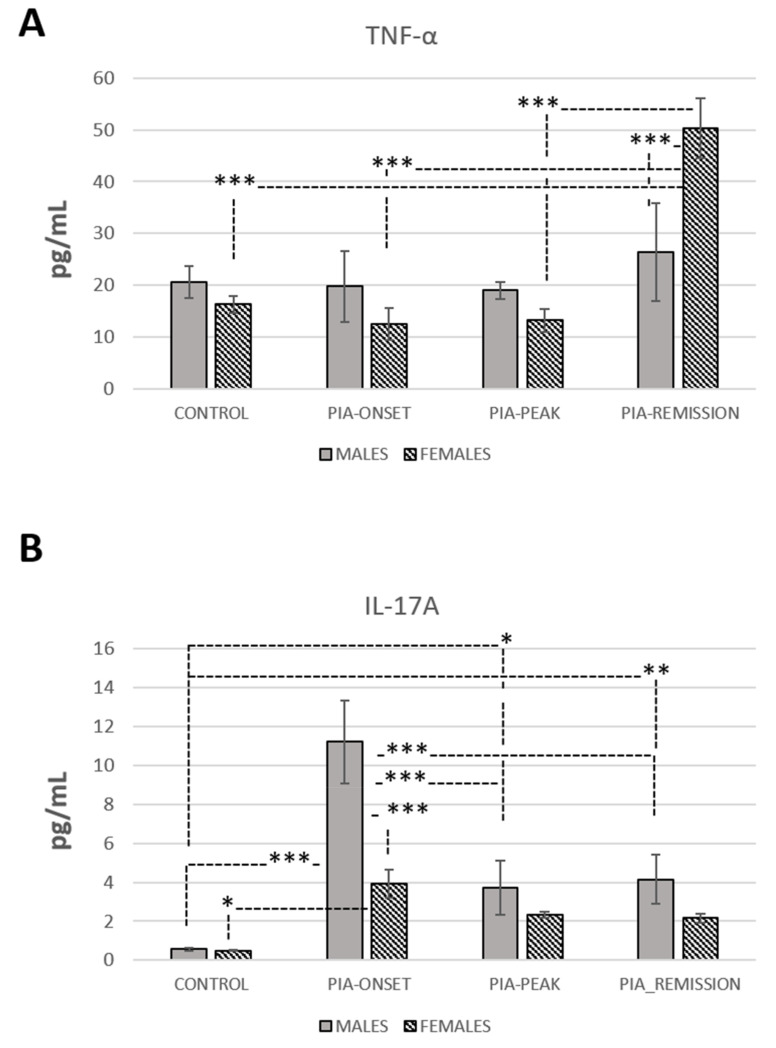
Expression profiles of TNF-α and IL-17A in serum during PIA. (**A**) Expression profile of TNF-α in serum. (**B**) Expression profile of IL-17A in serum. Both profiles are evaluated with ELISA in serum from male and female DA rats: control (treated with saline); PIA onset (between 9th and 12th day after induction); PIA peak (between 16th and 20th day after induction); PIA remission (between 20th and 25th day after induction). Values in pg/mL are expressed as mean ± SD (10 rats/group, *N* = 40). One-way ANOVA followed by the post hoc Scheffé test: * *p* < 0.05, ** *p* < 0.01, and *** *p* < 0.001.

These data suggest that BMP-4, IL-17A, and a lack of Noggin have a negative impact on adult neurogenesis in males and that IL-17A and a lack of Noggin have a negative impact on adult neurogenesis in females. BMP-7 expression in the hippocampus/dentate gyrus strongly correlates with TNF-α in females (r = 0.96, *p* < 0.001). TNF-α level in serum correlates with the number of DCX+ and Ki67+ cells in the dentate gyrus in females (r = 0.54, *p* = 0.040; r = 51, *p* = 0.048), which is the most pronounced in the remission phase of the disease (Figure 6 and Figure 7A). This could indicate the faster recovery of adult neurogenesis in females as well as a worsening of the clinical course of RA. However, further experiments are needed to confirm this. The overall Gremlin expression was at a very low level in both genders and groups.

### 3.7. Female Rats Show Higher Synovial Inflammation than Male Rats

To obtain the level of synovial inflammation, histopathological measures were performed using the Krenn synovitis scoring system (H&E staining) and T-cell staining (CD3). The data showed increased Krenn scores, i.e., low synovial inflammation at PIA onset and high-grade synovitis at PIA peak in both genders (Figure 6A). However, high-grade inflammation (Krenn score > 5) only persisted in female rats in the remission phase of the disease (Figure 6B). These findings are consistent with clinical observations (Figure 1) and a higher TNF-α level in the serum found in female rats (Figure 5A). In healthy rats, the synovial lining cell layer consisted of 1–2 layers of synoviocytes, with the synovial stroma containing mostly fat cells and blood vessels. Individual T lymphocytes (CD3^+^ cells) were sometimes found pericapillary (Figure 6A). At PIA onset, both in males and females, a notable increase of stromal cellularity was observed along with a slight proliferation of the synovial lining cell layer and underlining stromal fibrosis (Figure 6A). Inflammatory T cells (CD3^+^ cells) form small clusters near stromal capillaries and under the synovial lining cell layer. Female animals showed pannus formation at this time point. At PIA peak, the synovial lining cell layer became significantly proliferated in both genders. However, ulcerations and multinucleated cells were predominantly found in female synovial specimens.

This was followed by a significant increase in the absolute number of stromal cells due to a large buildup of inflammatory cells, predominantly T lymphocytes (CD3^+^ cells) in females (Figure 6A). Both genders had pannus formation at this time point. At PIA remission, a significant decrease in inflammation intensity was observed in male animals (Figure 6B), with a reduced number of synovial lining cell layers, stromal cellular density, and inflammatory infiltrate but with the presence of large multinucleated cells in the synovial stroma that was equal to low-mid grade synovitis (Figure 6A). In female animals, high-grade synovitis persisted even at PIA remission (Figure 6B), with high synovial lining cell layer proliferation (>four layers), high stromal cellular density due to persistent inflammatory infiltrate, pannus formation, and cartilage and bone destruction (Figure 6A). 

### 3.8. Female Rats Express a Higher Level of Synovial BMP-4 Than Male Rats during PIA

The profiling of the synovial BMP-4 protein in metatarsophalangeal joints by immunohistochemistry showed significant upregulation in female rats during PIA starting from disease onset, especially during peak and remission phases, in comparison with the findings in male rats (Figure 7B: PIA-peak *p* < 0.001; PIA-remission *p* < 0.001). In addition, the female rats had more proliferated synovial cells than male rats (Figure 6A, Figure 7A and Figure 8A). These findings are consistent with the clinical course and increase in TNF-α level in serum in females (Figure 1 and Figure 5A), especially in the remission phase (Figure 5A), which implies disease chronicity. The results are also correlated with increased IL-17A level in serum (Table 1: r = 0.53, *p* = 0.045) and Krenn score (Table 2: r = 0.87, *p* < 0.001). On the other hand, the IL-17A level in serum in females inversely correlates with BMP-4 and Noggin in the hippocampus/dentate gyrus simultaneously (Table 1). Weak Noggin expression was detected in normal synovial tissue in both genders, unlike BMP-4 expression (Figure 7A). In the PIA synovial tissue, Noggin expression was the most intense during PIA onset in both genders and was correlated with the IL-17A level in the serum (Table 2: r = 0.93, *p* < 0.001 in males and r = 0.83, *p* < 0.001 in females), but overall, there was greater Noggin expression in the females than there was in the males (Figure 7C), even in the control rats (*p* = 0.010). These results may imply greater BMP-4 inhibition by Noggin and thus attenuation of the anti-inflammatory role of BMP-4 during PIA despite its increased expression. Nevertheless, the greater expression of synovial BMP-4 and Noggin are consistent with a decreased level of adult hippocampal neurogenesis (Figure 2), hippocampal BMP-4 (Figure 3), and Noggin (Appendix A) expression in both genders. 

### 3.9. Female Rats Express a Higher Level of Synovial BMP-7 Than Male Rats during PIA

Profiling of synovial BMP-7 protein in metatarsophalangeal joints by immunohistochemistry showed significant upregulation in both genders during PIA with a gradual increase towards remission (Figure 8B). However, BMP-7 expression was higher in females with a significant increase in control and PIA-remission rats (Figure 8B: control *p* = 0.02; PIA-remission *p* < 0.001). These findings are consistent with the worse clinical course in females (Figure 1) and are correlated with an increased TNF-α level in the serum (Table 1: r = 0.76, *p* = 0.001) and in the Krenn score (Table 1: r = 0.71, *p* = 0.003) as well. The TNF-α level in the serum in the female rats correlates with BMP-7 in the hippocampus/dentate gyrus simultaneously (Table 1). Weak BMP-7 and Gremlin expression were detected in normal synovial tissue in both genders (Figure 8A) but were more significant in the females (*p* = 0.04; *p* < 0.001). Gremlin expression showed significant upregulation in both genders during PIA, with a gradual increase towards remission, but Gremlin levels were significantly higher in females (Figure 8C: PIA-onset *p* < 0.001; PIA-peak *p* < 0.001; PIA-remission *p* < 0.001). A correlation was found between synovial Gremlin expression and Krenn scores in both genders (Table 2: males r = 0.64, *p* = 0.015; females r = 0.63, *p* = 0.02) and between synovial Gremlin expression and IL-17A in serum, but only in females (Table 2: r = 0.78, *p* < 0.001). These results may imply greater BMP-7 inhibition by Gremlin, and thus attenuation of the anti-inflammatory role of BMP-7 during PIA despite its increased expression. Nevertheless, greater expression of synovial BMP-7 and Gremlin are consistent with a decreased level of adult hippocampal neurogenesis in both genders (Figure 2).

### 3.10. Correlation of IL-17A and TNF-α in Serum with Synovial Inflammation and Synovial BMPs, Noggin, and Gremlin

The synovial BMP-4 and BMP-7 correlate with the level of synovial inflammation (Krenn scores) in female rats (r = 0.87, *p* < 0.001; r = 0.71, *p* = 0.003). Synovial BMP-4 and BMP-7 and their inhibitor expressions are more upregulated in females than in males in general (Figure 6 and Figure 7). Furthermore, more significant correlations between IL-17A and TNF-α in the serum and synovial BMPs and their inhibitors were found in females than in males (Table 2). In addition, there was a significant inverse correlation of synovial BMP-4 with hippocampal BMP-4 (r = −0.79, *p* < 0.001) and hippocampal Noggin (r = −0.75, *p* = 0.001) and a positive correlation between the synovial BMP-7 and hippocampal BMP-7 (r = 0.71, *p* = 0.002) in females, unlike in males (data not shown in tables). However, data showed a significant inverse correlation between synovial Noggin and hippocampal Noggin in both genders (females: r = −0.71, *p* = 0.002; males: r = −0.83, *p* < 0.001) (data not shown in tables). These findings could explain the ultimate cause-and-effect relationship between the persistent synovial inflammation that causes systemic inflammation and inhibition of adult neurogenesis. However, the relationship of increased TNF-α in serum and the lowering of IL-17A in the remission phase of the disease in female rats should be elucidated more in further experiments, as should the increased expression of BMP-7 in both genders.

## 4. Discussion

In this study, we presented gender-related differences in the clinical course of pristine-induced arthritis and inflammatory-related changes in joints and adult neurogenesis in the hippocampus. Furthermore, we presented gender-related differences in expressions of BMP-4 and BMP-7 and their antagonists Noggin and Gremlin in the hippocampus and joints and the correlations between their expressions and TNF-α and IL-17A levels in the serum.

### 4.1. Clinical Course, Symptoms, and IL-17a and TNF-α Levels in Serum

Our results show that the onset of clinical signs in PIA developed earlier in male than in female rats (Figure 1). This is also consistent with an increase in IL-17A in the serum in males (Figure 5B); IL-17A was upregulated in females during PIA as well but at lower levels than in males. On the other hand, the clinical course in female rats worsened later in remission, i.e., showed higher clinical scores than in males (Figure 1). These findings in females are consistent with the inflammatory-related changes in synovial tissue of metatarsophalangeal joints (knee joints as well—data not shown) and high-grade inflammation (Krenn score > 5 in remission phase), along with the TNF-α in the serum. This is not surprising since worse disease activity and function are described in female human patients with RA compared to men [25].

Furthermore, it is well known that TNF-α governs chronicity in multiple diseases, including RA [33]. TNF-α is one of the main pro-inflammatory cytokines that mediates inflammation and bone degradation in RA [34]. In addition, TNF-α, together with IL-17A, initiates the RA pathology. On the other hand, IL-17A has a significant, if not central, role in the pathogenesis of RA [35]. It acts directly on osteoblasts and synovial fibroblasts to promote inflammation [36]. However, IL-17A acting alone has relatively modest effects on different immune cell types, and when combined with TNF-α, it massively increases the production of other cytokines, i.e., IL-6 and IL-8, and consequently increases the level of inflammation [37]. In addition, the synergy between IL-17A and TNF-α occurs only when IL-17A acts before TNF-α and not when TNF-α acts first [38,39]. This pattern can be seen in our study as well (Figure 5), i.e., IL-17A increases first, especially in males, and afterwards in remission, TNF-α increases in both genders, especially in females, with a significant difference. This concept is important when applied to clinical conditions, such as arthritis, in which there is no need for large amounts of cytokines because of these synergistic interactions. Locally, this is further amplified by cell–cell interactions, specifically between T cells and stromal cells, such as synoviocytes [40,41]. Regarding the local features described above, it is important to consider that cytokines interact as a team in blood and tissues, including the brain and hippocampal neurogenic niche.

Given the higher prevalence of RA in women, we suspect that such differences in cytokine expressions are due to sexual hormones, specifically estrogens, which are strong modulators of immune response and function that are associated with RA.

Immuno-enhancing estrogens stimulate the transcription factors (e.g., NF-κB) that are important for producing pro-inflammatory cytokines, i.e., TNF-α, and thus promoting synovitis [42,43].

Why TNF-α increased more in females than in males in the remission phase of our study is not entirely clear. It could be associated with estrogen regulation, or on the other hand, it could be associated with lower immunosuppressive androgen concentrations (i.e., testosterone, androstenedione, and dehydroepiandrosterone sulfate) in the serum as well as in the synovial fluid [44]. Accordingly, it is known that testosterone reduces the serum level of TNF-α and attenuates the inflammatory process [45,46]. Thus, the protective role of androgens could be one of the reasons for better clinical and synovial histopathological outcome in male rats from our study. This needs further confirmation and clarification in future research. 

The administration of estrogens may also have a protective effect on RA development by delaying the disease onset [47]. A similar effect can be seen in the onset of PIA in our study. This is positive effect, unlike the one seen in the remission phase of the disease, where the females show a prolongation of the symptoms. It is already known that the activity of several estrogens (E1, E2, E3, and E4) is different and depends not only on the hormone itself but also on the specific disease [43]. Furthermore, estrogens act directly on the immune system through α and β estrogen receptors (ERα and ERβ), which have a distinct affinity to estrogen concentrations and modifications and can affect RA in a dose-dependent manner [44]. This might be due to variations in the distribution of ERs in immune cells.

Further experiments are needed to determine exactly what type and what dose of estrogen is responsible for these different effects. Furthermore, it was documented that estrogen regulates interleukin-17-producing T helper cells (Th17) in experimental autoimmune arthritis by increasing the number of Th17 cells in lymph nodes during the early phase of arthritis development and by decreasing it in joints during established arthritis [47].

This is very similar to the dynamic of serum IL-17A levels in female rats from our study. However, little information exists on serum IL-17A during RA and its association with estrogen and androgen levels. We are not sure why male rats express a higher level of serum IL-17A in our study, especially at the onset of PIA. This should be investigated in future experiments.

### 4.2. BMPs in Synovium

BMP signalling nor gender differences in BMP expressions in RA are entirely described. Synovial BMPs increased in both genders in the joints, but the females generally showed higher expressions of BMPs and their antagonists (Noggin and Gremlin). These findings are consistent with a higher level of inflammation in female rats. As already described [48], BMP signalling reduces the expression of pro-inflammatory and pro-destructive factors in both unstimulated and stimulated RA synoviocytes, mainly with the IL-17A and TNF-α combination, so BMP signalling could have an anti-inflammatory role in the control and maintenance of low levels of pro-inflammatory factors in healthy synoviocytes and probably also in the early stages of RA. Thus, BMPs play a disease-controlling role as joint-protective factors [49], maintaining a quiescent phenotype of the synovial lining layer [50]. However, stimulation of RA synoviocytes induces expression not only of synovial BMP ligands but also of BMP antagonists (e.g., Noggin and Gremlin) as well, mainly after treatment with TNF-α alone or in combination with IL-17A [48]. These findings suggest that in the late stages of RA, the BMP signalling pathway probably can no longer control and maintain the low levels of pro-inflammatory factors, which then rises and persists at chronically high levels, contributing to RA pathogenesis [51]. Thus, the high levels of the BMP ligands produced by stimulated synoviocytes could also contribute to the chronic inflammation associated with RA by acting in a paracrine fashion on the surrounding cells that are present in the inflamed synovium. Thus, when produced at higher levels, BMPs have been shown to induce a pro-inflammatory phenotype in endothelial cells [52,53] and to stimulate chemotactic responses in monocytes/macrophages [54,55], which play a central role in RA. 

Furthermore, BMP pathway activation increases monocyte adherence to endothelial cells [52], stimulates the production of pro-inflammatory cytokines, including TNF-α, IL-1, and IL-6, by macrophages [56,57], induces maturation of dendritic cells [58], increases T-cell proliferation and activation [59,60], and promotes T helper 17 (Th17) differentiation [60]. In addition, BMPs are chemotactic for mesenchymal stem cells [61,62,63] and thus stimulate osteoclast differentiation and activity [64], which could contribute to the bone destruction seen in RA [49]. On the other hand, in the study by Bramlage et al., BMP-4 was expressed in normal synovial tissue, but decreased expression in patients with RA was found in response to increased TNF-α [15]. In our study, we found the expression of synovial BMP-4 in the control rats, both in the males and females (Figure 7B), which is consistent with the findings mentioned above. However, a decrease in synovial BMP-4 expression can only be seen in male rats compared to female rats (Figure 7B). It was shown that synovial BMP-7 increases in patients with RA and that the levels correlate with the severity of the disease [48]. We can see the same pattern in our study, with a significantly greater expression of BMP-7 in females, even in control rats (Figure 8B). 

In addition, synovial BMP antagonists, Noggin and Gremlin, increased in both genders, probably to inhibit the inflammation-related increase of BMPs. Again, we found their significantly higher expression in females (Figure 7C and Figure 8C). In females, synovial expressions of BMP-4 and both antagonists (Gremlin and Noggin) were significantly positively correlated with IL-17A, and the synovial expression of BMP-7 significantly positively correlated with TNF-α (Table 2). These findings could imply the importance of IL-17A in initiating the increase of synovial expression of BMP-4, Gremlin and Noggin, and TNF-α in initiating the increase of expression of BMP-7, but mostly in females.

### 4.3. BMPs in Hippocampus/Dentate Gyrus

As previously mentioned, if the joint BMPs are inhibited, the production of pro-inflammatory and pro-destructive factors by RA synoviocytes is increased [48], which could further contribute, we assume, to systemic and neuronal inflammation and adult neurogenesis deterioration. Many studies confirmed that BMP-4 contributes to a decline in adult neurogenesis, which indicates that the inhibition of BMP signalling is a mechanism for the rapid expansion of the pool of new neurons in the adult hippocampus [22]. In our study, hippocampal BMP-4 was only upregulated in males at the PIA peak. However, later in remission, BMP-4 was downregulated in both genders (Figure 3), which correlates with the increase in the number of proliferating (Ki67^+^) and neural progenitor (SOX-2^+^) cells, immature neurons (DCX^+^), and their maturation speed and survival (ratio of DCX^+^ to Ki67^+^ cells), but mostly in females (Figure 2). The increase of hippocampal BMP-4 in males during PIA peak is not the main reason for the decline in adult neurogenesis. The expression of Noggin was downregulated with the onset of PIA in both genders and in the remission (Appendix A), which is another reason for the decline in neurogenesis. In addition, Noggin is in negative correlation with IL-17A in serum, which suggests that IL-17A not only causes decline in adult neurogenesis [14] but also through the downregulation of Noggin [14,18].

These data suggest that BMP-4, IL-17A, and a lack of Noggin have a negative impact on adult neurogenesis in males and that IL-17A and a lack of Noggin have a negative impact on adult neurogenesis in females. It is already documented that Noggin has a beneficial impact on adult neurogenesis [18], but it is possible that Noggin production is not only inhibited by IL-17A [14]. This could cause less Noggin production by T cells in arthritis, which could lead to the unresponsiveness of rat PBMCs to TNF-α [65]. On the other hand, TNF-α has a positive correlation with the number of DCX^+^ and Ki67^+^ cells in the dentate gyrus in females, which may indicate a faster recovery of adult neurogenesis [65] but at the same time, may worsen the clinical course of arthritis [33]. TNF-α is a negative regulator of progenitor proliferation in adult hippocampal neurogenesis and acts synergistically with IL-17A to shape different CNS inflammation [66]. On the contrary, several studies have shown that TNF-α influences the proliferation, survival, and differentiation of neural progenitor cells [67,68]. In our results, the remission phase began earlier in male rats, and the female rats processed the remission phase with a higher level of clinical scores.

However, the remission in females is correlated with the number of proliferating cells and maturation of immature neurons (Table 1), and it is consistent with the much higher TNF-α level in serum (Figure 5A and Table 1). Further experimental work should explore if this increase is a beneficial factor for the recovery of adult neurogenesis or only a predictor for the arthritis chronicity. Data also showed that untreated control male rats possessed a higher level of BMP-4 and its antagonist Noggin (Appendix A) than female rats, which correlates with the larger number of neuroblasts/immature neurons (DCX^+^ cells) and the ratio of DCX^+^ to Ki-67^+^ cells (DCX/Ki67), reflecting a higher basic level of hippocampal proliferation as well as newly born cell maturation speed and survival. These results may imply that there is a higher level of initial protection for adult neurogenesis in male rats. In addition, there was a significant inverse correlation of synovial BMP-4 with hippocampal BMP-4 and hippocampal Noggin and a positive correlation between synovial BMP-7 and hippocampal BMP-7 in females, which is unlike what was seen in the male rates (data not shown in tables). However, the data showed a significant inverse correlation between synovial Noggin and hippocampal Noggin in both genders (data not shown in tables). These findings could explain the ultimate cause-and-effect relationship between the persistent synovial inflammation that causes systemic inflammation and a negative effect on adult neurogenesis [48]. However, further experiments should further elucidate the relationship between increased TNF-α in the serum and the lowering of IL-17A in the remission phase of the disease in female rats as well as the increased level of BMP-7 in both genders. Furthermore, it is interesting that the DCX^+^ cells in the untreated control male rats co-express significantly more BMP-4 and Noggin ligands than females (Appendix A). These results may imply a higher level of protection of adult neurogenesis in general in male rats. In our results, hippocampal BMP-7 was mainly upregulated in males during PIA, although in remission, it is upregulated in both genders (Figure 4). This is in strong correlation with TNF-α. BMP-7 is a multifunctional cytokine with demonstrated neuroregeneratory and protective potential, especially against TNF-α [69]. This upregulation in remission could be a protective response against TNF-α increase. Furthermore, the DCX^+^ cells in the hippocampus/dentate gyrus in male rats, in general, co-express more BMP-7 and Gremlin (Appendix A). ligands than in female rats These results may indicate a gender-related protective feature that provides neuroprotection during RA. As already said, in our study, the female rats showed an increase of adult neurogenesis in the remission phase compared to the male rats, although this recovery of adult neurogenesis in remission could be a result of decreased BMP-4 expression during the entirety of PIA (onset, peak, and remission) (Figure 3B). In addition, we cannot claim that these newly born neuroblasts will survive and differentiate into adult neurons. For this, further investigation is required.

## 5. Conclusions

Our data show that gender differences exist in BMP expressions in the periphery–hippocampus axis and IL-17A and TNF-α levels in serum, which could imply differences in mechanisms for the onset and progression of the disease, clinical course severity, and subsequent neurological complications between genders (Table 3). These findings could explain the ultimate cause-and-effect relationship between the persistent synovial inflammation causing systemic inflammation and the negative effect on adult hippocampal neurogenesis. However, further experiments should continue to elucidate the relationship between increased TNF-α in the serum, the lowering of IL-17A, and faster neurogenesis recovery in the remission phase in females as well as the increased level of hippocampal BMP-7 in both genders during remission.

## Figures and Tables

**Figure 6 ijms-22-12163-f006:**
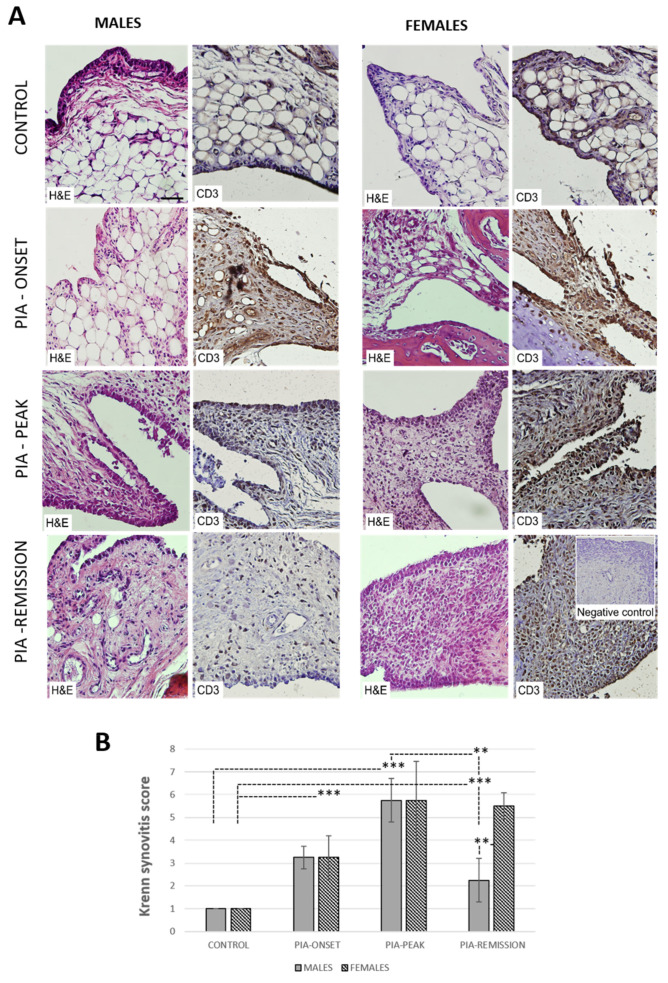
Pristane-induced arthritis causes greater inflammation in synovial tissue in female than in male rats. (**A**) Representative photomicrographs show staining with H&E for measuring the level of inflammation (Krenn scoring system) and anti-CD3 antibody for detecting T cells in paraffin-embedded sections of the metatarsophalangeal joints obtained from male and female DA rats: control (treated with saline); PIA onset (between 9th and 12th day after induction); PIA peak (between 16th and 20th day after induction); PIA remission (between 20th and 25th day after induction). CD3 is expressed in synovial and T cells. (**B**) Krenn synovitis score of metatarsophalangeal joints. Synovitis was scored by two independent observers under light microscope (3 slides/rat × 6 rats/group). Values are expressed as mean value ± SD (*N* = 24). One-way ANOVA followed by the post hoc Scheffé test: ** *p* < 0.01 and *** *p* < 0.001. Insert shows staining in a section incubated without primary anti-CD3 antibody (negative control). Scale bars indicate 50 μm.

**Figure 7 ijms-22-12163-f007:**
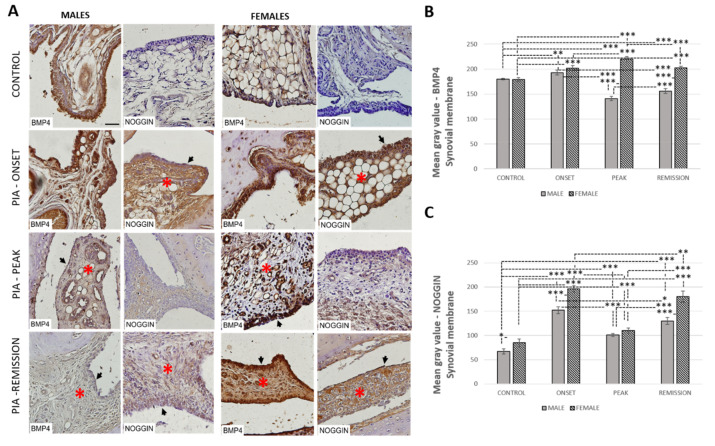
Pristane-induced arthritis upregulates BMP-4 and Noggin expression in synovial tissue in female rats more than in male rats. (**A**) Representative immunohistochemical pictures show staining with anti-BMP-4 and anti-Noggin antibodies in paraffin-embedded sections of metatarsophalangeal joints obtained from male and female DA rats: control (treated with saline); PIA onset (between 9th and 12th day after induction); PIA peak (between 16th and 20th day after induction); PIA remission (between 20th and 25th day after induction). BMP-4 is expressed in stromal cells and synovial lining cells. *gender differences in immunohistochemical staining of stroma; arrows: gender differences in immunohistochemical staining and proliferation of synovial lining cell layer. (**B**,**C**) BMP-4 and Noggin immunoreactivity in synovial tissue of metatarsophalangeal joints. Immunohistochemical staining quantification was performed using Cell F v3.1 software analysis on 12 regions of interest (3 slides/rat × 6 rats/group). Values are expressed as mean grey value ± SD (*N* = 24). One-way ANOVA followed by the post hoc Scheffé test: * *p* < 0.05, ** *p* < 0.01 and *** *p* < 0.001. Scale bars indicate 50 μm.

**Figure 8 ijms-22-12163-f008:**
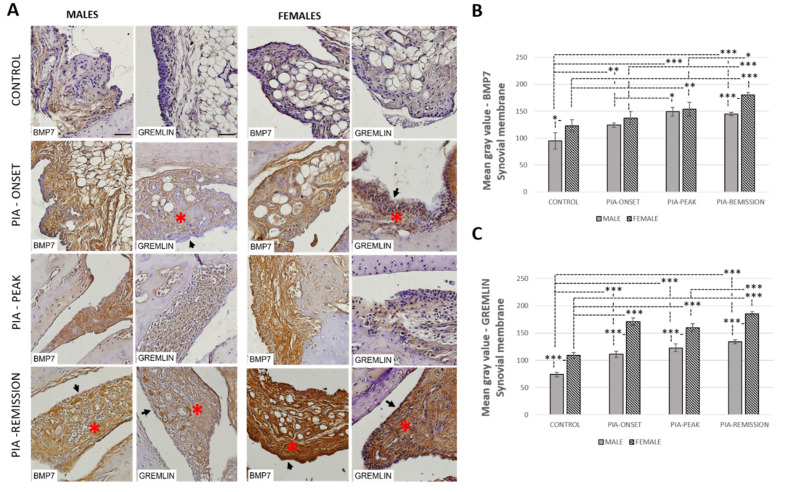
Pristane-induced arthritis upregulates BMP-7 and Gremlin expression in synovial tissue in female rats more than in male rats. (**A**) Representative immunohistochemical pictures show staining with anti-BMP-7 and anti-Gremlin antibodies in paraffin-embedded sections of metatarsophalangeal joints obtained from male and female DA rats: control (treated with saline); PIA onset (between 9th and 12th day after induction); PIA peak (between 16th and 20th day after induction); PIA remission (between 20th and 25th day after induction). BMP-7 is expressed in stromal cells and synovial lining cells. *gender differences in immunohistochemical staining of stroma; arrows: gender differences in immunohistochemical staining of synovial lining cell layer. (**B**,**C**) BMP-7 and Gremlin immunoreactivity in synovial tissue of metatarsophalangeal joints. Immunohistochemical staining quantification was performed using Cell F v3.1 software analysis on 12 regions of interest (3 slides/rat × 6 rats/group). Values are expressed as mean grey value ± SD (*N* = 24). One-way ANOVA followed by the post hoc Scheffé test: * *p* < 0.05, ** *p* < 0.01 and *** *p* < 0.001. Scale bars indicate 50 μm.

**Table 2 ijms-22-12163-t002:** Pearson’s correlation between synovial expressions of BMPs, Noggin, and Gremlin, IL-17A and TNF-α levels in serum, and Krenn synovitis score.

Variable	Gender	Average Grey Value BMP4	Average Grey Value BMP7	Average Grey Value Noggin	Average Grey Value Gremlin
TNF-α	Males	−0.06	0.22	0.04	0.29
	Females	0.03	0.76 **	0.38	0.46
IL-17A	Males	0.45	0.34	0.93 ***	0.35
	Females	0.53 *	0.31	0.83 ***	0.78 ***
Krenn	Males	−0.45	0.69 **	0.30	0.64 *
	Females	0.87 ***	0.71 **	0.23	0.63 *

* *p* < 0.05, ** *p* < 0.01 and *** *p* < 0.001.

**Table 3 ijms-22-12163-t003:** Gender-related differences during PIA.

Variable	Gender					
	Males			Females		
	PIA-Onset	PIA-Peak	PIA-Remission	PIA-Onset	PIA-Peak	PIA-Remission
DCX^+^ cells	↓	↓	↓	↓	↓	↔
Ki67^+^ cells	↔	↓	↓	↓	↓	↔
DCX/Ki67	↓	↓	↓	↔	↓	↑
TNF-α	↔	↔	↔	↔	↔	↑↑
IL-17A	↑↑	↑	↑	↑	↑	↔
Hippocampal BMP4	↓	↑	↓	↓	↓	↓
Hippocampal BMP7	↓	↓	↑	↔	↔	↑
Hippocampal Noggin	↓	↓	↓	↓	↓	↓
Hippocampal Gremlin	↑↑	↔	↔	↔	↔	↔
Synovial BMP4	↑	↓	↓	↑	↑↑	↑↑
Synovial BMP7	↑	↑	↑	↑	↑	↑↑
Synovial Noggin	↑	↑	↑	↑↑	↑	↑↑
Synovial Gremlin	↑	↑	↑	↑↑	↑↑	↑↑
Krenn synovitis score	↑	↑	↑	↑	↑	↑↑

Arrows interpretation: ↔ without changes compared with control; ↑ increase compared with control; ↓ decrease compared with control; ↑↑ pronounced increase compared with the other gender.

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
