# Peer review of "Gender-Related Differences in BMP Expression and Adult Hippocampal Neurogenesis within Joint-Hippocampal Axis in a Rat Model of Rheumatoid Arthritis"

_ijms, 2021, doi:10.3390/ijms222212163_

Round 1

Reviewer 1 Report

The work of Omrcen et al. studied the effect that BMP-4 and BMP-7, expressed in synovial tissue, plays on systemic inflammation and adult neurogenesis. The study focuses on determining the characteristics adopted by this phenomenon according to gender. The study was developed in an animal model of arthritis induced after pristane injection of Dark Agouti rats. On one hand, the authors observed gender differences of BMP expression levels in the periphery-hippocampus axis. On the other hand, the authors found differences in IL-17A and TNF-alpha levels in the serum of induced animals which responded to the gender difference. The authors conclude that the results obtained could imply differences in the mechanisms of disease onset and progression.

Materials and Methods Section:

The authors show the protocol followed to perform immunofluorescence in paraffin-embedded tissue sections. Due to the nature of the sample, this type of staining is often complicated.

-Is it correct that authors use 1% BSA for the blocking step? Isn't this concentration too low?

-Did the authors adapt the immunofluorescence staining protocol based in paraffin-embedded tissue section to avoid artifacts like lipofuscin-like autofluorescence? This is no mentioned in the manuscript.

Results and Discussion Sections:

-In addition of serum results, it would be interesting to extend the study to the infiltrates of IL17-producing T cells in synovial tissue. These data could help to better understand the differences observed as a function of gender in the chosen experimental model of pristane-induced arthritis.

-How the authors relate IL17A / TNF-alpha production with the hormonal profile associated to each gender during the development of the experimental disease model? Is this connection relevant in the conclusions obtained in the work? The authors should adequately discuss this point.

-In the context of the work, the authors discuss that IL17-A acting alone has modest effects on the immune system cells. On the contrary, when IL17-A is combined with TNF-alpha, the production of other cytokines such as IL-6 and IL-8 is massively increased. Did the authors study IL-6 and IL-8 secretion in the proposed experimental model? What are the results obtained?

-Have the authors developed a detailed study of the chronicity of the induced disease in female rats (beyond day 30)? If so, which parameters have been considered exactly?

-Authors state in the manuscript that “The onset of clinical signs in male PIA rats was more progressive than in female rats”. According to Figure 1 data, the symptom curves for both genders have a similar slope. Would be more accurate to say that male animals have an earlier onset of symptoms after PIA induction.

-Which is the objective of showing, in various locations of the article, a p = 0.000? Wouldn't it be more convenient to show the specific numerical value in each case using the appropriate scientific notation?

Author Response

We couldn’t thank you enough for the beneficial suggestions and for taking all the effort to make our Manuscript better: Gender-related differences in BMP expression and adult hippocampal neurogenesis within joint-hippocampal axis in a rat model of rheumatoid arthritis”.

The responses and explanations related to the comments are listed below and highlighted in the Revised Manuscript:

Reviewer 1   -   Comments and Suggestions to Authors:

The work of Omrcen et al. studied the effect that BMP-4 and BMP-7, expressed in synovial tissue, plays on systemic inflammation and adult neurogenesis. The study focuses on determining the characteristics adopted by this phenomenon according to gender. The study was developed in an animal model of arthritis induced after pristane injection of Dark Agouti rats. On one hand, the authors observed gender differences of BMP expression levels in the periphery-hippocampus axis. On the other hand, the authors found differences in IL-17A and TNF-alpha levels in the serum of induced animals which responded to the gender difference. The authors conclude that the results obtained could imply differences in the mechanisms of disease onset and progression.

 Materials and Methods Section:

The authors show the protocol followed to perform immunofluorescence in paraffin-embedded tissue sections. Due to the nature of the sample, this type of staining is often complicated.

-Is it correct that authors use 1% BSA for the blocking step? Isn't this concentration too low?

-Did the authors adapt the immunofluorescence staining protocol based in paraffin-embedded tissue section to avoid artifacts like lipofuscin-like autofluorescence? This is no mentioned in the manuscript.

Response to the Reviewer:

We thank the reviewer for the constructive comment. We followed the procedure according to the manufacturer's instructions. However, to make sure there would be no unspecific staining, we performed the negative control staining (tissue samples were treated with an identical procedure under the same conditions, but with the omission of polyclonal primary antibodies. We inserted this explanation into text (Page 5) and into figure legends (Figure 2, 3, 4 and 6, as well in Supplement figures 3 and 4).

Results and Discussion Sections:

-In addition of serum results, it would be interesting to extend the study to the infiltrates of IL17-producing T cells in synovial tissue. These data could help to better understand the differences observed as a function of gender in the chosen experimental model of pristane-induced arthritis.

Response to the Reviewer:

We thank the reviewer for the logical suggestion. Indeed, we have in mind to expand our research and in addition to explore synovial Th17 cells. 

-How the authors relate IL17A / TNF-alpha production with the hormonal profile associated to each gender during the development of the experimental disease model? Is this connection relevant in the conclusions obtained in the work? The authors should adequately discuss this point.

Response to the Reviewer:

We added additional text in the Discussion part (Page 20).

-In the context of the work, the authors discuss that IL17-A acting alone has modest effects on the immune system cells. On the contrary, when IL17-A is combined with TNF-alpha, the production of other cytokines such as IL-6 and IL-8 is massively increased. Did the authors study IL-6 and IL-8 secretion in the proposed experimental model? What are the results obtained?

Response to the Reviewer:

We thank the reviewer for good observation. In this study, we didn’t analyzed IL-6 and IL-8 we only relied on the reference number 30. However, when we expand our future research, this will one of the goals.

-Have the authors developed a detailed study of the chronicity of the induced disease in female rats (beyond day 30)? If so, which parameters have been considered exactly?

Response to the Reviewer:

We thank the reviewer for the question. We exactly wanted to develop a detailed study of the chronicity beyond day 30, but it was not possible because of Ethics Committee counseling, which made a limitation of days for agonizing the laboratory animals.

-Authors state in the manuscript that “The onset of clinical signs in male PIA rats was more progressive than in female rats”. According to Figure 1 data, the symptom curves for both genders have a similar slope. Would be more accurate to say that male animals have an earlier onset of symptoms after PIA induction.

Response to the Reviewer:

We changed the sentences (Abstract, Page 7 in Results, and Page 19 in Discussion part).

-Which is the objective of showing, in various locations of the article, a p = 0.000? Wouldn't it be more convenient to show the specific numerical value in each case using the appropriate scientific notation?

Response to the Reviewer:

We adopted and wrote as p < 0.001.

We couldn’t thank you enough for the beneficial suggestions and for taking all the effort to make our Manuscript better: Gender-related differences in BMP expression and adult hippocampal neurogenesis within joint-hippocampal axis in a rat model of rheumatoid arthritis”.

The responses and explanations related to the comments are listed below and highlighted in the Revised Manuscript:

Reviewer 1   -   Comments and Suggestions to Authors:

The work of Omrcen et al. studied the effect that BMP-4 and BMP-7, expressed in synovial tissue, plays on systemic inflammation and adult neurogenesis. The study focuses on determining the characteristics adopted by this phenomenon according to gender. The study was developed in an animal model of arthritis induced after pristane injection of Dark Agouti rats. On one hand, the authors observed gender differences of BMP expression levels in the periphery-hippocampus axis. On the other hand, the authors found differences in IL-17A and TNF-alpha levels in the serum of induced animals which responded to the gender difference. The authors conclude that the results obtained could imply differences in the mechanisms of disease onset and progression.

 Materials and Methods Section:

The authors show the protocol followed to perform immunofluorescence in paraffin-embedded tissue sections. Due to the nature of the sample, this type of staining is often complicated.

-Is it correct that authors use 1% BSA for the blocking step? Isn't this concentration too low?

-Did the authors adapt the immunofluorescence staining protocol based in paraffin-embedded tissue section to avoid artifacts like lipofuscin-like autofluorescence? This is no mentioned in the manuscript.

Response to the Reviewer:

We thank the reviewer for the constructive comment. We followed the procedure according to the manufacturer's instructions. However, to make sure there would be no unspecific staining, we performed the negative control staining (tissue samples were treated with an identical procedure under the same conditions, but with the omission of polyclonal primary antibodies. We inserted this explanation into text (Page 5) and into figure legends (Figure 2, 3, 4 and 6, as well in Supplement figures 3 and 4).

Results and Discussion Sections:

-In addition of serum results, it would be interesting to extend the study to the infiltrates of IL17-producing T cells in synovial tissue. These data could help to better understand the differences observed as a function of gender in the chosen experimental model of pristane-induced arthritis.

Response to the Reviewer:

We thank the reviewer for the logical suggestion. Indeed, we have in mind to expand our research and in addition to explore synovial Th17 cells. 

-How the authors relate IL17A / TNF-alpha production with the hormonal profile associated to each gender during the development of the experimental disease model? Is this connection relevant in the conclusions obtained in the work? The authors should adequately discuss this point.

Response to the Reviewer:

We added additional text in the Discussion part (Page 20).

-In the context of the work, the authors discuss that IL17-A acting alone has modest effects on the immune system cells. On the contrary, when IL17-A is combined with TNF-alpha, the production of other cytokines such as IL-6 and IL-8 is massively increased. Did the authors study IL-6 and IL-8 secretion in the proposed experimental model? What are the results obtained?

Response to the Reviewer:

We thank the reviewer for good observation. In this study, we didn’t analyzed IL-6 and IL-8 we only relied on the reference number 30. However, when we expand our future research, this will one of the goals.

-Have the authors developed a detailed study of the chronicity of the induced disease in female rats (beyond day 30)? If so, which parameters have been considered exactly?

Response to the Reviewer:

We thank the reviewer for the question. We exactly wanted to develop a detailed study of the chronicity beyond day 30, but it was not possible because of Ethics Committee counseling, which made a limitation of days for agonizing the laboratory animals.

-Authors state in the manuscript that “The onset of clinical signs in male PIA rats was more progressive than in female rats”. According to Figure 1 data, the symptom curves for both genders have a similar slope. Would be more accurate to say that male animals have an earlier onset of symptoms after PIA induction.

Response to the Reviewer:

We changed the sentences (Abstract, Page 7 in Results, and Page 19 in Discussion part).

-Which is the objective of showing, in various locations of the article, a p = 0.000? Wouldn't it be more convenient to show the specific numerical value in each case using the appropriate scientific notation?

Response to the Reviewer:

We adopted and wrote as p < 0.001.

Reviewer 2 Report

In this study, the authors investigate the correlation of experimental arthritis, hippocampal neurogenesis, and factors regulating both conditions. While the study is novel and of interest, some points should be addressed before publication can be recommended. In general, the authors show correlations of various metrics which is hard to follow in the text. I strongly suggest to focus on the most relevant findings and showing the results of the entire study either as a supplement (in case the journal allows) or, summarized in a table at the end of the results section. Beyond, the following specific points need to be addressed:

  • The structure of the abstract is diffuse and should be changed: Rational, hypothesis, methods, results and conclusion.
  • The relationship between neuropsychiatric comorbidities and RA should be better outlined in the introduction section and a hypothesis should be stated.
  • Please specify how decalcification was performed.
  • Please state the pH of the citrate buffer used during HIER
  • Please specify how anesthesia was performed during transcardial perfusion
  • Please provide order and/or RIDD numbers for the used primary antibodies.
  • Please describe more in detail how the specificity of the immunohistological studies was verified. Did the authors use isotype controls in parallel experiments? Beyond, with the provided magnification, nothing can be said about the specificity of the immunohistochemical stains. For example, the anti-CD3 stains show a high background, specific stains should be membranous.
  • The authors state that “PIA was induced in genetically susceptible DA rats with an incidence of 99.9% in 244 both genders.” Please give the precise numbers such as 99 out of 100 rats.
  • The labelings KI67 in figure 2 are hard to see, please change the color. Beyond, the name of the antibody should be given always at the very same place of an image (for example lower, left).
  • Scale bars should be shown only once per figure in case these are the same magnifications.
  • Figure 2 shows various aspects of hippocampal cellular responses upon RA induction. The authors should try to explain more in detail what the results are and provide a brief interpretation. For example, what does it mean that in females at PIA-Onset numbers of DCX+ and DCX/Ki67+ cells decreases, whereas the overall number of KI67+ cells increases?
  • P=0.000 is not existing as it would mean that the data are 100% significant. Please adopt.
  • The authors are sometimes too strong with their conclusions. For example, they state the following: These findings all together confirm the negative impact of IL-17A on adult neurogenesis in both genders, although a lower level of BMP-4 in females may indicate a faster recovery of adult neurogenesis”. Since no functional studies have been performed nothing can be concluded about the relevance of IL17A on adult neurogenesis. Please adopt.

Author Response

We couldn’t thank you enough for the beneficial suggestions and for taking all the effort to make our Manuscript better: Gender-related differences in BMP expression and adult hippocampal neurogenesis within joint-hippocampal axis in a rat model of rheumatoid arthritis”.

The responses and explanations related to the comments are listed below and highlighted in the Revised Manuscript:

Reviewer 2   -   Comments and Suggestions to Authors:

In this study, the authors investigate the correlation of experimental arthritis, hippocampal neurogenesis, and factors regulating both conditions. While the study is novel and of interest, some points should be addressed before publication can be recommended. In general, the authors show correlations of various metrics which is hard to follow in the text. I strongly suggest to focus on the most relevant findings and showing the results of the entire study either as a supplement (in case the journal allows) or, summarized in a table at the end of the results section.

Response to the Reviewer:

We thank the reviewer for the constructive comments to make our paper better. We removed Noggin and Gremlin hippocampal expressions from the manuscript and inserted them into Supplement file. Other expressions (hippocampal and synovial BMP expressions, as well as synovial Noggin and Gremlin expressions) showed significant changes, so we decides these results to leave in the manuscript. We summarized all results together in Table on Page 23.

Please find our other responses to the comments below.

Beyond, the following specific points need to be addressed:

The structure of the abstract is diffuse and should be changed: Rational, hypothesis, methods, results and conclusion.

Response to the Reviewer:

We agree with the reviewer and have separated parts accordingly, although the journal is not recommended it, so we hope it will be accepted.

The relationship between neuropsychiatric comorbidities and RA should be better outlined in the introduction section and a hypothesis should be stated.

Response to the Reviewer:

We added additional text in introduction part (Page 2).

Please specify how decalcification was performed.

Response to the Reviewer:

We added new explanation (Page 4 under “Hind paws”).

Please state the pH of the citrate buffer used during HIER

Response to the Reviewer:

We added pH of citrate buffer (Page 4).

Please specify how anesthesia was performed during transcardial perfusion

Response to the Reviewer:

We described it on Page 4 under “Experimental Design”. We added references with previously described procedure.

Please provide order and/or RIDD numbers for the used primary antibodies.

Response to the Reviewer:

We added RIDD numbers as suggested (Page 5).

Please describe more in detail how the specificity of the immunohistological studies was verified. Did the authors use isotype controls in parallel experiments? Beyond, with the provided magnification, nothing can be said about the specificity of the immunohistochemical stains. For example, the anti-CD3 stains show a high background, specific stains should be membranous.

Response to the Reviewer:

We thank the reviewer for the kind suggestion. We added an explanation in text (Page5), as well and the negative controls in Figure 6.

The authors state that “PIA was induced in genetically susceptible DA rats with an incidence of 99.9% in 244 both genders.” Please give the precise numbers such as 99 out of 100 rats.

Response to the Reviewer:

We added a precise number (Page 6).

The labelings KI67 in figure 2 are hard to see, please change the color. Beyond, the name of the antibody should be given always at the very same place of an image (for example lower, left).

Response to the Reviewer: We change the presentation of Ki67 figures in Figure2, to better see the red fluorescence of Ki67 positive cells. As well, we inserted the name of the antibody in the left lower corner.

Scale bars should be shown only once per figure in case these are the same magnifications.

Response to the Reviewer:

We left only one scale bar per figure.

Figure 2 shows various aspects of hippocampal cellular responses upon RA induction. The authors should try to explain more in detail what the results are and provide a brief interpretation. For example, what does it mean that in females at PIA-Onset numbers of DCX+ and DCX/Ki67+ cells decreases, whereas the overall number of KI67+ cells increases?

Response to the Reviewer:

We suppose that reviewer thought about the changes in males, not in females. Accordingly, we inserted a brief explanation in results part (Page 7), since we already discussed this issue in Discussion part of the manuscript.

P=0.000 is not existing as it would mean that the data are 100% significant. Please adopt.

Response to the Reviewer:

We adopted and wrote as p < 0.001.

The authors are sometimes too strong with their conclusions. For example, they state the following: These findings all together confirm the negative impact of IL-17A on adult neurogenesis in both genders, although a lower level of BMP-4 in females may indicate a faster recovery of adult neurogenesis”. Since no functional studies have been performed nothing can be concluded about the relevance of IL17A on adult neurogenesis. Please adopt.

Response to the Reviewer:

We adopted this sentence (Page 12).

Round 2

Reviewer 1 Report

The authors resolved some issues raised by the reviewer. The authors indicate that the rest of the questions that are not resolved now will be addressed in future work. It is highly recommended materializing this proposal since it would give it the strength that the study showed here needs.